# *TNFα* rs1800629 Polymorphism and Response to Anti-*TNFα* Treatment in Behçet Syndrome: Data from an Italian Cohort Study

**DOI:** 10.3390/jpm13091347

**Published:** 2023-08-31

**Authors:** Maria Carmela Padula, Angela Anna Padula, Salvatore D’Angelo, Nancy Lascaro, Rosa Paola Radice, Giuseppe Martelli, Pietro Leccese

**Affiliations:** 1Rheumatology Department of Lucania and Rheumatology Institute of Lucania (IReL), San Carlo Hospital of Potenza, 85100 Potenza, Italy; 2Department of Science, University of Basilicata, 85100 Potenza, Italy

**Keywords:** Behçet syndrome, pharmacogenetics, polymorphism, TNF-α

## Abstract

Tumor Necrosis Factor-alpha (*TNFα*) rs1800629 (-308G>A) is a single nucleotide polymorphism (SNP) related to variable responses to anti-*TNFα* therapy. This therapy is efficient in severe and refractory manifestation of Behçet syndrome (BS), an auto-inflammatory systemic vasculitis. We investigated (1) the association between rs1800629 genotypes and responses to therapy and (2) the correlation between SNP and clinical patterns in a cohort of 74 BS Italian patients receiving anti-*TNFα* therapy with a follow-up of at least 12 months. The rs1800629 was genotyped through amplification, direct sequencing and bioinformatics analyses. The rs1800629 GG and GA genotypes were assessed as predictors of outcomes dividing the patients between therapy responders and non-responders. The rs1800629 GG and GA genotypes were found, respectively, in 59/74 (79.7%) and 15/74 BS patients (21.3%) (*p* < 0.05). We identified 16/74 (21.9%) non-responder patients, of which 9/16 (56.3%) showed the GG genotype and 7/16 (43.7%) the GA genotype. A total of 50/58 (86.2%) responder patients showed the GG genotype, and 8/58 (13.8%) the GA genotype (*p* < 0.05). The percentage of non-responder females (68.8%) was significantly higher than non-responder males (31.2%) (*p* < 0.05). No correlation between SNP and clinical patterns was observed. To successfully include rs1800629 as a predictive biomarker of *TNFα* inhibitor response, genome-wide association studies in larger, well-characterised cohorts are required.

## 1. Introduction

Tumor Necrosis Factor-alpha (*TNFα*) is a pleiotropic cytokine involved in the regulation of a wide spectrum of biological processes, including cell proliferation, differentiation, apoptosis, lipid metabolism, and coagulation. This cytokine has been implicated in a variety of diseases, including autoimmune diseases, insulin resistance, cancer, and also in the current pandemic coronavirus infectious disease (COVID-19) [1,2,3,4,5]. Its critical role has also been reported in the pathogenesis of several systemic rheumatic diseases, including Behçet syndrome (BS), a chronic vasculitis characterized by partially unknown etiology and wide clinical heterogeneity [2,6,7,8,9,10,11]. BS clinical signs range from mucocutaneous, articular, vascular, and gastrointestinal involvement, to ocular and neurologic involvement. These clinical manifestations may co-exist in the same patient, defining a broad spectrum of clinical phenotypes [6,7,8,9,10,11]. Both genetics and environmental factors are involved in disease etiopathogenesis [2,7,8,10,11,12,13,14]. Human leukocyte antigen (HLA)-B*51 was the most strongly associated genetic marker of BS [10,11,12,13,14]. Other non-HLA genetic loci were identified within genes mainly involved in the inflammation and immunity processes. In fact, the associations with BS susceptibility were demonstrated for several genes, such as *Endoplasmic reticulum aminopeptidase 1 (ERAP1)* and *Interleukin 10 (IL10)* and *IL23R-IL12RB2*, as well as *TNFα* [2,10,13,14].

The heterogeneity of treatments reflects the variability of clinical signs. The 2018 update of the EULAR recommendations for the management of BS provided to physicians evidence-based recommendations for patient care in single disease manifestations. Due to BS being characterized by a relapsing and remitting course, the treatment objective is to counter the inflammatory response and to prevent organ damage [15]. Colchicine, azathioprine (AZA), thalidomide, interferon-alpha (IFN α), cyclosporine-A, cyclophosphamide, glucocorticoids, apremilast, and the blockage of *TNFα*, IL-1, and IL-6 IL-17 are the main significant treatments reported [15]. The need for tailoring the treatments to patient phenotype (characterized by various clinical manifestations) rather than having a single involvement was summarized in the review by Bettiol and colleagues [9]. The authors reported the treatment for the three major BS phenotypes: patients with the “mucocutaneous and articular” phenotype should be treated with colchicine, alone or in combination with corticosteroids; AZA can be used in case of resistance or intolerance to colchicine; and anti-*TNFα* or IFNα should be considered for refractory or severe forms. In case of an “extra-parenchymal and peripheral vascular phenotype”, immunosuppressant drugs and additional anticoagulants in selected patients should be recommended. Traditional immunosuppressants (mainly AZA) should be the first-line treatment, while anti-*TNFα* therapy could be a second-line treatment. In the case of the “parenchymal neurological and ocular phenotype”, AZA is recommended as a first-line treatment after an induction therapy with high-dose steroids. In the case of severe manifestations or intolerance to AZA, anti-*TNFα* drugs should be employed [9].

The approach of targeting *TNFα* has considerably improved success in the treatment of BS, in particular for patients with refractory, severe BS, and in particular for ocular, central nervous system, and gastrointestinal manifestations, as well as vascular involvement [6,9,15,16,17,18]. Five different *TNFα* inhibitors are now used: infliximab, etanercept, adalimumab, golimumab, and certolizumab-pegol. Infliximab is the most commonly administered drug. The blockade of *TNFα* is characterized by a high clinical efficacy and rapid onset of action, but the lack of response is a common trait after repeated infusions [4,5,6]. Although biological treatments with anti-*TNFα* agents are effective in BS, not all patients are definite responders. Non-responder patterns could be due to alternative non-*TNFα*-related pathways of inflammation, the presence of anti-drug antibodies or development and polymorphic alleles of the *TNFα* gene [4,5,6,7,17,18,19,20,21,22,23,24,25]. *TNFα* (RefSeq NG_007462.1) is located quite close to the Major Histocompatibility Complex (MHC) at chromosome 6 (6p21.33), and encodes a 233-amino-acid type II transmembrane protein. *TNFα* is formed by 4 exons and about 3000 nucleotides, including both 5′UTR and 3′UTR regulatory regions. *TNFα* -308G>A (rs1800629; NG_007462.1:g.4682G>A; HGVS nomenclature) was classified as a drug-response polymorphism in several specific databases (for example dbSNP, ClinVar, ClinGen Allele Registry). We previously studied the genotype distribution of BS patients compared with healthy controls and we found a statistically significantly higher frequency of the *TNFα* rs1800629 GA genotype in patients than in controls. No significant association was recognized between the polymorphism and the clinical parameters, as well as between SNP and disease severity [2]. The association between the SNP and both susceptibility and clinical patterns of BS was investigated with conflicting data [4,5,6,7,17,18,19,20,21,22,23,24,25]. Poorer data are currently available regarding the association of this polymorphism and its responsiveness to *TNFα* blockers in BS patients, in particular in Italian populations. The aim of this study was to investigate (1) the association of rs1800629 with the response to anti-*TNFα* therapy (outcome) in a cohort of Italian patients with BS, and (2) the correlation between SNP and clinical manifestations.

## 2. Materials and Methods

This is a retrospective, monocentric cohort study conducted at the Rheumatology Institute of Lucania/San Carlo Hospital (Potenza, Italy). The Regional Ethics Committee approved this study (Permit Number: 705/2017).

### 2.1. Patient Enrolment

Patients with BS were identified from our large database. Inclusion criteria were BS patients fulfilling the ISG criteria [26] who received anti-*TNFα* therapy with a follow-up of at least 12 months after drug initiation, who had available medical records (with information on treatment duration and efficacy), and who consented to participate. Prior to enrolling, all subjects provided their written, informed consent. Demographic and clinical data of enrolled patients were collected from medical records and analysed. The rs1800629 genotype was assessed as a predictor of outcomes, dividing the patients in two groups: therapy responders and non-responders. We set a period of 12 months as the time interval useful for evaluating clinical response and we distinguished the primary (patients who do not respond to the induction therapy) and secondary loss of response (patients who respond to the therapy after an induction regimen, but subsequently lose response during maintenance treatment). Potential confounders could be age, gender, ethnicity, and clinical manifestations.

### 2.2. Genotyping

Genomic DNA was extracted from whole blood by using a commercial kit (Nuclear Laser Medicine Srl, Settala, Italy) according to the manufacturer’s recommendations. After isolation, DNA was quantified using the NanoDrop™ 1000 spectrophotometer (NanoDrop Technologies, Inc., Wilmington, Delaware) to assess the quality and concentration of the nucleic acid for downstream application. In vitro PCR was performed using home-made specific primer pairs for rs1800629 coverage (primer design by NCBI Primer-Blast tool) (forward: 5′TTCCCTCCAACCCCGTTTTC3′, reverse: 5′CTGCACCTTCTGTCTCGGTT3′). PCR amplification was carried out using Q5 Hot Start High-Fidelity DNA Polymerase (New England BioLabs Inc., Ipswich, USA), according to the manufacturer’s recommendations: 5 uL of 5X Reaction Buffer was mixed with 0.5 uL of 10 mM dNTPs, 1.25 uL of each primer, and 0.25 uL of High-Fidelity DNA polymerase, and added to each sample for a final volume of 25 uL. The amplification program was (1) 98 °C for 5 min (initial denaturation); (2) 94 °C for 1 min, 58 °C for 1 min, 72 °C for 2 min (thermocycling, repeated 35 times); and (3) 72 °C for 7 min (final extension). A negative control was also used in PCR reactions. Amplification products were analysed using gel electrophoresis (1.5% agarose gel) and sequenced with the Microsynth sequencing service (Sanger method). In silico analysis was performed downstream using the NCBI-BlastN similarity search tool and Mutation Surveyor 3.9 software.

### 2.3. Statistics

Descriptive and analytical statistics were carried out using SPSS Statistic version 17 for Windows (SPSS Inc., Chicago, IL, USA). Demographic and genotype frequencies and clinical manifestations were compared using a chi-square goodness of fit test or Fisher’s exact test (for 2 × 2 tables). A *p*-value less than 0.05 was considered statistically significant.

## 3. Results

A total of 74 BS patients (44 males, 30 females; mean age: 43.1 ± 11.3 years) meeting the inclusion criteria were enrolled. Patients’ predominant lesions were oral aphthous (100%), skin lesions (82.4%), eye involvement (75.7%), and genital ulcers (50.0%). HLA-B*51 was found in the 67.6% of cases.

### Response to TNFα Therapies and Pharmacogenetics

We compared the patients for therapy responses to anti-*TNFα* drugs, and we found 58/74 (78.4%) therapy responders and 16/74 (21.6%) non-responder patients. Within the non-responder group, 3/16 (18.75%) patients were primary non-responders and 13/16 (81.25%) were characterized by losing drug efficacy during the treatment (secondary loss of response). No statistically significant differences were found when the responder and non-responder groups were analysed for the presence/absence of each clinical manifestation (*p* > 0.05) (Table 1).

For genetic characterization, the overall analysis showed that the *TNFα* rs1800629 wild-type GG genotype was recognized in 59/74 (79.7%) BS patients; the heterozygous genotype (GA) was identified in 15/74 (21.3%) patients (*p* < 0.05). We identified that 50/58 (86.2%) responders showed the GG genotype and 8/58 (13.8%) responder patients showed the GA genotype, while 9/16 (56.25%) non-responder patients showed the GG genotype and 7/16 (43.75%) the GA genotype (*p* = 0.008; OR: 4.86 (1.41–16.76) (Table 2).

The two groups were also analysed for gender differences and stratified for anti-*TNFα* agent (Table 3). The percentage of non-responder females (68.8%; 11/16 patients) was significantly higher than non-responder males (31.2%; 5/16 patients) (*p* = 0.009, OR: 0.22 (0.07–0.83). We found that 60/74 (81.8%) patients were treated with Infliximab, 12/74 (16.2%) with Adalimumab, 1/74 (1.35%) with Certiluzumab Pergol and 1/74 (1.35%) with Golimumab. We also performed a sub-analysis of responses based on each anti-*TNFα* drug. We found 45/58 (77.6%) responders treated with Infliximab and 15/16 (93.75%) non-responder patients receiving the same therapy (*p* = 0.1439, OR: 0.23 (0.03–1.92)). A total of 11/58 (19.0%) patients treated with Adalimumab were responder patients, while 1/16 (6.25%) was a non-responder (*p* > 0.05). No differences were found when responders and non-responders were compared for Certulizumab Pergol and Golimumab (*p* > 0.05).

## 4. Discussion

We studied the association between *TNFα* rs1800629 genotypes and anti-*TNFα* therapy responses, as well as between the same SNP and the clinical response in BS. We found that the frequency of the rs1800629 wild-type GG genotype was statistically significantly higher in the case of therapy response compared to the GA genotype, suggesting a possible role of the SNP-containing genotype in affecting the drug response. We also recognized a higher frequency of non-responder females compared with males. No significant differences were found when responders and non-responders were stratified for all anti-*TNFα* drugs and compared.

BS is an auto-inflammatory systemic vasculitis characterized by a relapsing-remitting course. A wide spectrum of pharmaceutical agents is now available for treating clinical manifestations of the disease; the therapy goal is to prevent symptom worsening triggered and supported by different molecular and cellular mechanisms, such as inflammation [4,5,6,9,15,27]. Although BS etiopathogenesis remains partially unknown, it has been reported that hexogen triggers (such as infective agents and unbalanced microbiomes) are responsible for immune activation resulting in inflammatory symptoms in genetically predisposed subjects [2,4,5,6,7,8,9,10,11,12,13,14,15,27]. A dysregulated Th1/Th2 and Th17/Treg cells balance and a hyper-activation of pro-inflammatory cytokines (in particular IL-1, IL-6, and *TNFα*) are associated with chronic inflammation and typical BS clinical phenotypes [4,5,6,9,27]. In this scenario, *TNFα* inhibitors are significant effective therapeutic tools, due to anti-*TNFα* targeting being a good way to achieve disease remission. Not all patients have long-term remission, so the discontinuation of the *TNFα* inhibitor therapy has become an area of interest, due to obvious economic and risk-benefit concerns. These aspects underline the need to have useful biomarkers for predicting the response to therapy [4,5,6,9,16,18,27,28]. Due to advances in DNA genotyping and sequencing approaches, genetic variations become an interesting source of markers able to predict treatment responses and to obtain useful information within the field of personalized medicine. Literature data about *TNFα* rs1800629 have shown a consistent amount of data about the role of SNP in BS susceptibility, while limited and conflicting data are available on the ability to predict anti-*TNFα* treatment responses in BS based on gene polymorphisms. The association between *TNFα* rs1800629 SNP and BS susceptibility is a common literature finding in several ethnic groups [7,8,17,18,19,20,21,22,23,24,25], as well as in Italian populations, based on our recently published data [2]. The ability of SNP to affect *TNFα* expression and the inflammation pathway was also reported in relation to the SNP localization; that is, within the gene regulatory region and, in particular, within the gene promoter [7,17,19,20,21,22,23,24,25].

In the present investigation, the distribution of rs1800629 genotypes showed a higher percentage of non-responder patients carrying the A allele (GA genotype) in agreement with previous data about the SNP role in influencing drug response. These studies reported that patients with inflammatory diseases having the AA genotype may be less likely to achieve improvements in clinical manifestations compared with patients carrying the GG genotype when treated with anti-*TNFα* drugs [4,20,24,29,30,31,32,33,34,35,36,37,38,39,40].

Gender differences in anti-*TNFα* therapy responses have not previously been investigated. We found a higher frequency of non-responder females compared with non-responder males, and this difference was statistically significant, suggesting that genetics could be one of the possible contributors modulating therapy responses in the sexes, together with anatomical, physiological, neuronal, hormonal, psychological and social factors. A previous study by our group investigated gender differences in BS clinical manifestations and found that BS tends to be less aggressive in female patients [41].

In this study, we also performed an explorative analysis on the distribution of non-responder patients when our cohort was stratified for anti-*TNFα* drugs, and we found a lower response in the case of Infliximab, also depending on the higher number of patients treated with this drug. The role of genetic factors for the lack of therapeutic response to anti-*TNFα* agents was previously suggested in patients with rheumatoid arthritis (RA), psoriatic arthritis (PsA), ankylosing spondylitis (AS), juvenile idiopathic arthritis (JIA), and Sjögren’s syndrome, as well as in inflammatory bowel disease, in particular Crohn’s disease (CD), but also in Wegener’s granulomatosis and sarcoidosis [4,29,30,31,32,33,34,35,36,37,38,39,40].

The drugs showed significant differences in structure and in clinical efficacy, as well as in their mechanisms of action. Firstly, TNF-α antagonists could be divided into two main types of agents: monoclonal antibodies and soluble receptors. Infliximab, Adalimumab, and Golimumab are biological drugs against human *TNFα*, while Etanercept is direct against human *TNFα* receptors. Infliximab is a chimeric human-mouse anti-*TNFα* monoclonal antibody, while Adalimumab and Golimumab are fully humanized anti-*TNFα* antibodies. Etanercept is formed from the extracellular portion of the two human TNF-R2 linked to the Fc portion of human IgG1, while Certolizumab is a fragment of an anti-TNF-α IgG1 monoclonal antibody. Infliximab, Adalimumab, and Etanercept are more effective for the treatment of RA, PsA, and AS. This variability response could be related to differences in pharmacokinetics, tissue distribution and functional properties of each anti-*TNFα* agent [1,9,16,18,27,29,40,42,43,44,45,46,47,48,49,50,51,52].

Although different mechanisms of action have been suggested, such as pro-inflammatory cytokine down-regulation, apoptosis induction, complement-dependent cytotoxicity, and antibody-dependent cell-mediated cytotoxicity, the cellular and molecular mechanisms of action of the anti-*TNFα* antibodies remain partially unknown, and the mechanisms of the lack of drug response are also partially unclear [1,4,5,9,16,18,27,29,40,42,43,44,45,46,47,48,49,50,51,52]. In both contexts, pharmacogenetics and pharmacogenomics represent a good way to compose the puzzle by finding the missing tiles.

## 5. Conclusions

Pharmacogenetics and pharmacogenomics represent the new frontiers for the discovery of potential genetic markers of biological responses to *TNFα* inhibitors. The identification of robust and validated SNP-based biomarker panels is one of the most significant goals of Predictive, Preventive and Personalised Medicine (PPPM). In particular, insights into how the therapy response is genetics-related could lead to the development of strategies for predicting the non-response to anti-*TNFα* therapy prior to treatment initiation and for implementing a stratified and tailored approach of clinical care. This paper could provide a first step in this direction in a cohort of Italian patients with BS. Since our sample size is relatively small, analyses of a larger cohort of patients are needed to confirm the study findings and to explain the SNP role as an outcome predictor; also, in randomized clinical trials, it is useful to include gene variation in clinical decision-making and drug management.

## Figures and Tables

**Table 1 jpm-13-01347-t001:** Clinical manifestations and demographic characteristics of BS patients divided into responders and non-responders.

	Responders (*n* = 58)	Non-Responders (*n =* 16)	*p*-Value	OR	95% CI
Clinical Manifestations					
Oral ulcers					
with	58 (100.0%)	16 (100.0%)	0.7585	3.63	0.00–NA
without	0 (0.0%)	0 (0.0%)			
Genital ulcers					
with (*n* = 37)	26 (44.8%)	11 (68.8%)	0.0902	0.37	0.11–1.20
without (*n* = 37)	32 (55.2%)	5 (31.2%)			
Papulopustular lesions					
with (*n* = 51)	42 (72.4%)	9 (56.3%)	0.2162	2.04	0.65–6.41
without (*n* = 23)	16 (27.6%)	7 (43.7%)			
Erythema nodosum					
with	23 (39.7%)	5 (31.2%)	0.5394	1.45	0.44–4.71
without	35 (60.3%)	11 (68.8%)			
Follicolitis					
with	8 (13.8%)	2 (12.5%)	0.8934	1.12	0.21–5.88
without	50 (86.2%)	14 (87.5%)			
Anterior uveitis					
with	20 (34.5%)	6 (37.5%)	0.8229	0.88	0.28–2.76
without	38 (65.5%)	10 (62.5%)			
Posterior uveitis					
with	34 (58.6%)	8 (50.0%)	0.5378	1.42	0.47–4.30
without	24 (41.4%)	8 (50.0%)			
Arthritis					
with	14 (24.1%)	2 (12.5%)	0.3168	2.23	0.45–11.2
without	44 (75.9%)	14 (87.5%)			
CNS involvement					
with	15 (25.9%)	4 (25.0%)	0.9443	1.05	0.29–3.75
without	43 (74.1%)	12 (75.0%)			
Superficial venous thrombosis					
with	6 (10.3%)	2 (12.5%)	0.8058	0.81	0.15–4.45
without	52 (89.7%)	14 (87.5%)			
Deep venous thrombosis					
with	4 (6.9%)	2 (12.5%)	0.4672	0.52	0.09–3.13
without	54 (93.1%)	14 (87.5%)			
GI involvement					
with	14 (24.1%)	2 (12.5%)	0.9433	0.95	0.27–3.44
without	44 (75.9%)	14 (87.5%)			

Abbreviations: OR, odds ratio; CI, confidence interval.

**Table 2 jpm-13-01347-t002:** Genotype of rs1800629 for responders and non-responders in BS patients.

Genotypes	Responders (*n =* 58)	Non-Responders (*n =* 16)	*p*-Value	OR (95% CI)
GG (*n =* 59)	50 (86.2%)	9 (56.3%)	0.008 *	4.86 (1.41–16.76)
GA (*n =* 15)	8 (13.8%)	7 (43.7%)		

Abbreviations: OR, odds ratio; CI, confidence interval; * statistically significant.

**Table 3 jpm-13-01347-t003:** Demographic features of BS patients and responses to all anti-*TNFα* drugs in responder and non-responder groups.

	Total (*n =* 74)	Responders (*n =* 58)	Non-Responders (*n =* 16)	*p*-Value
Demographics				
Female	30 (40.5%)	19 (32.8%)	11 (68.8%)	0.009 *
Male	44 (59.5%)	39 (67.2%)	5 (31.2%)	
Age	43.1 ± 11.3	42.8 ± 12.3	45.6 ± 10.2	
Anti-*TNFα* drugs				
Infliximab	60 (81.8%)	45 (77.6%)	15 (93.75%)	0.1439
Adalimumab	12 (16.2%)	11 (19.0%)	1 (6.25%)	0.1957
Certiluzumab Pergol	1 (1.35%)	1 (1.7%)	0 (0.0%)	0.3494
Golimumab	1 (1.35%)	1 (1.7%)	0 (0.0%)	0.3494

* statistically significant.

## Data Availability

The data presented in this study are available on request from the corresponding author.

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
