# Peer review of "TNFα rs1800629 Polymorphism and Response to Anti-TNFα Treatment in Behçet Syndrome: Data from an Italian Cohort Study"

_jpm, 2023, doi:10.3390/jpm13091347_

Round 1
Reviewer 1 Report
|
REVIEW CRITERIA |
SCALE |
Line/ Sentence
|
Comments |
||
|
|
|
|
|||
|
Title |
poor major revision minor revision good |
Title |
Use colon (:)
|
||
|
Abstract |
poor major revision minor revision good |
Abstract para |
Abstract needs more clarity towards the Behcet syndrome and importance of TNFα for treatment. |
||
|
Introduction |
poor major revision minor revision good |
Nil |
The abstract should be extended to provide a more comprehensive overview of Behçet's syndrome and incorporate prior research endeavors aimed at its treatment. |
||
|
Materials and Methods |
poor major revision minor revision good |
1. Line no. 80-81 2. Nil 3. Line no. 97 |
1. Provide statement/ reference or any proof for the standard procedure followed for extraction of Genomic DNA from whole blood
2. Provide the instrumentation condition, calibration and other scientific parameters while performing this work.
3. Also mention the version and licence no. of. SPSS software use for the statistical calculations
|
||
|
Results |
poor major revision minor revision good |
|
|
||
|
Discussion |
poor major revision minor revision good |
Nil |
Although the main cause of Behcet syndrome is still unknown, the author must still provide a valid statement regarding the significance and relevance of their work..
|
||
|
Conclusion |
poor major revision minor revision good |
|
|
||
|
References |
poor major revision minor revision good |
|
|
||
|
Data representation |
poor major revision minor revision good |
|
|
||
|
Graphs and figures |
NA |
|
|
||
Minor editing of the English language required
Author Response
Thank you for the possibility to improve tha qulity of our manuscripit. Our point-by-point responses are the following
Title: use colon (:)
Response 1: We modified the manuscript according to your suggestions.
Abstract: Abstract needs more clarity towards the Behcet syndrome and importance of TNFα for treatment.
Reponse 2: The sentence “This therapy is efficient in severe and refractory manifestation of Behçet syndrome (BS), an auto-inflammatory systemic vasculitis” was added according to your suggestion.
Introduction: The abstract should be extended to provide a more comprehensive overview of Behçet's syndrome and incorporate prior research endeavors aimed at its treatment.
Response 3: Both aspects were added within the Introduction section from line 35 to line 66.
Materials and methods:
Provide statement/ reference or any proof for the standard procedure followed for extraction of Genomic DNA from whole blood.134.
Response 4.1: We use a commercial kit and we added this information at line 113-114.
Provide the instrumentation condition, calibration and other scientific parameters while performing this work.
Response 4.2: According to our opinion and work flow, the methods reported all information about this issue (use of standard procedures, assessment of DNA quality, the presence of negative control in PCR reaction). Can you suggest any info to add please?
Also mention the version and licence no. of. SPSS software use for the statistical calculations
Response 4.3: The software version number was added at line 131; we use a free trial version.
Discussions: Although the main cause of Behcet syndrome is still unknown, the author must still provide a valid statement regarding the significance and relevance of their work.
Response 5: We underlined the role of inflammation in BS etiopathogenesis and the significance of target inflammation pathway (in particular TNF pro-inflammatory cytokine) from line 181 to 192. The conclusion section also underlines the significance of our work in the field of Predictive, Preventive and Personalised Medicine: “This paper could provide a first step in this direction in a cohort of Italian patients with BS” (line 258-259).
Reviewer 2 Report
Here are few correction
In the abstract line no. 13 authors started sentence with with the letter 74 you can not start English sentence like this you can write like a total of 74 patients etc.
In same way do not start sentence with rs number. you can use The
Please remove such things from other parts of article as well.
I wrote above
Author Response
Thank you for the possibility to improve tha qulity of our manuscripit. We modified the manuscript according to your suggestions.
Reviewer 3 Report
The paper titled "TNFα rs1800629 Polymorphism and Response to Anti-TNFα Treatment in Behçet Syndrome: Data from an Italian Cohort Study" investigates the association between the TNFα rs1800629 (GG and GA genotypes) polymorphism and the response to anti-TNFα therapy in BS patients within an Italian cohort. The authors examined the correlation between rs1800629 genotypes and treatment response, as well as its relationship with clinical patterns.
The method section would benefit from a clearer structure, with subsections delineating patient cohort details, statistics, and genotyping assays.
Overall, the study contributes valuable insights into the role of TNFα rs1800629 polymorphism in anti-TNFα treatment response among BS patients. And restructuring the method section would enhance the presentation of the study's design.
Author Response
Thank you for the possibility to improve tha qulity of our manuscripit. The method section has been updated by adding subsections, according to your suggestions.